# Elucidation of the Relationship between Intrinsic Viscosity and Molecular Weight of Cellulose Dissolved in Tetra-N-Butyl Ammonium Hydroxide/Dimethyl Sulfoxide

**DOI:** 10.3390/polym11101605

**Published:** 2019-10-01

**Authors:** Daqin Bu, Xiangzhou Hu, Zhijie Yang, Xue Yang, Wei Wei, Man Jiang, Zuowan Zhou, Ahsan Zaman

**Affiliations:** Key Laboratory of Advanced Technologies of Materials (Ministry of Education), School of Materials Science and Engineering, Southwest Jiaotong University, Chengdu 610031, China; budaqin619102@my.swjtu.edu.cn (D.B.); xzHu@my.swjtu.edu.cn (X.H.); yangzhijie@my.swjtu.edu.cn (Z.Y.); yangxue@my.swjtu.edu.cn (X.Y.); tzweir@swjtu.edu.cn (W.W.); zwzhou@swjtu.edu.cn (Z.Z.); zaman@my.swjtu.edu.cn (A.Z.)

**Keywords:** cellulose, TBAH/DMSO aqueous solution, intrinsic viscosity, molecular weight

## Abstract

The determination of molecular weight of natural cellulose remains a challenge nowadays, due to the difficulty in dissolving cellulose. In this work, tetra-n-butylammonium hydroxide (TBAH) and dimethyl sulfoxide (DMSO) aqueous solution (THDS) were used to dissolve cellulose in a few minutes under room temperature into true molecular solutions. That is to say, the cellulose was dissolved in the solution in molecular level, and the viscosity of the solution is linearly dependent on the concentration of cellulose. The relationship between the molecular weight of cellulose and the intrinsic viscosity tested in such dilute solutions has been established in the form of the Mark–Houwink equation, $\left[\eta \right]=0.24\times {\mathrm{DP}}^{1.21}$. The value of 1.21 indicates that the cellulose molecules dissolve in THDS quite well. The cellulose dispersion in the THDS was proved to be in molecular level by atomic force microscope (AFM) and dynamic light scattering (DLS). The reliability of the established Mark–Houwink equation was cross-checked by the gel permeation chromatography (GPC) and traditional copper (II) ethylenediamine (CED) method. No considerate degradation was observed by comparing the intrinsic viscosity and the degree of polymerization (DP) values of the original with and the regenerated cellulose samples. The natural cellulose can be molecularly dispersed in the multiple-component solvent (THDS), and kept stable for a certain period. A time efficient and reliable method has been supplied for determination of the degree of polymerization and the molecular weight of cellulose.

## 1. Introduction

Cellulose is the most abundant natural polymer in the Earth, which is widely produced by various plants, bacteria, and algae [1]. The increasingly serious environmental and energy issues have prompted the development of natural polymer materials [2,3]. Cellulose are widely used in our daily life and modern industrial production as renewable materials [4], composite materials [5], and cellulose derivatives [6,7,8]. The molecular weight (M) of cellulose affects the processing conditions, as well as the performance of the final cellulose products. The accurate determination of molecular weight is one of the key issues for the development of cellulose-based materials in laboratory research, as well as industry. There are three methods for determination of molecular weight of cellulose. One is the GPC method [9,10,11], which usually conducted for cellulose derivatives. The other is the viscosity method [9,12]. The static light scattering (SLS) [13] method is also used for determination of the cellulose molecular weight. The premise of the methods is that cellulose is dissolved in the solvent without degradation. The commonly-used method to measure the molecular weight of cellulose in the laboratory is the cuprammonium hydroxide (Cuoxam) method (GB/T 9170–1999). Although the test process of the Cuoxam method is simple and fast, cellulose degrades greatly in Cuoxam solution. There is another common copper (II) ethylenediamine (CED) method (GB/T 1548–2004), which can accurately measure the molecular weight of cellulose, but it is difficult to prepare CED solution. Therefore the search for new efficient solvents to measure the molecular weight of cellulose has always been a research hotspot. However, Cellulose can hardly be dissolved in common solvents, due to its extensive intra- and intermolecular hydrogen bonding [14,15,16,17], and the amphiphilicity distribution of the crystal surfaces [18,19,20]. As is well known, most of the solvent systems have various deficiencies. For example, C_2_S/NaOH used in viscose fiber production, with sulfuric acid as coagulation bath, causes serious pollution to the air and water, and it is not suitable for testing the molecular weight of cellulose. Although side reaction occurs during the dissolving process [21], N-methylmorpholine-N-oxide (NMMO) [22,23,24] has been prevalently applied in the cellulose wet spinning industry to take the place of traditional C_2_S/NaOH viscose fiber production. Cellulose can be dissolved in NMMO at 90 °C, and at high temperatures the relationship between molecular weight and intrinsic viscosity of cellulose has become so complicated that no one has studied it. Many new solvents have been developed and the relationship between intrinsic viscosity and molecular weight of cellulose has been studied, such as lithium chloride/N,N-dimethylacetamide (LiCl/DMAc), $\left[\eta \right]=(1.278\times 10^{-4}){M_{w}}^{0.83}$ [25,26], this equation is applicable to cellulose with a molecular weight range of 12.5 × 10^4^–70.0 × 10^4^, and cannot characterize cellulose with a smaller molecular weight. BmimAc/DMSO, $\left[\eta \right]=2.5\times {\mathrm{DP}}^{0.83}$ [9], belongs to the class of ionic liquids (ILs) [17]; LiOH, $\left[\eta \right]=(2.7\times 10^{2})M^{0.79}$ [27,28] and paraformaldehyde (PF)/DMSO [23,24] $\left[\eta \right]=3.01\times {\mathrm{DP}}^{0.81}$. The alkaline solvents have been studied extensively [29,30,31,32], in which, NaOH/urea [30,31,32] is a quite unique aqueous solvent for fast dissolution of cellulose involving the breakage of the hydrogen bond of cellulose, and dissolving into stable solution with balanced intra- and intermolecular action among the cellulose and the solvent constituents. The relationship between the intrinsic viscosity and molecular weight of cellulose in 6 wt % NaOH/4 wt % urea aqueous was first proposed by Zhang et al., which was $\left[\eta \right]=\left(2.45\times 10^{2}\right)M^{0.815}$ [33]. The work also indicated that NaOH solution causes cellulose degradation, and the degradation rate is slower than that of Cuoxam solution. In order to develop a new solvent system for the determination of molecular weight of cellulose, a lot of research had been done on addition of co-solvent. Recently, the addition of co-solvent has gained increasing interest in improving the dissolving capacity for cellulose, such as DMSO [34,35], urea [36], and choline chloride [37]. However, it is still a challenge to quickly dissolve natural cellulose with high degree of polymerization (DP) at near room temperature [38]. Tetra-n-butylammonium hydroxide (TBAH) with DMSO has been proved to be an efficient multiple-component solvent for dissolving cellulose [39,40,41]. Even natural cellulose can be completely dissolved in several minutes under room temperature [4,42,43]. These reports inspire us to establish a fast and accurate method based on viscosity to determine the molecular weight of cellulose, which is anticipated to be much more convenient than the commonly used copper (II) ethylenediamine (CED) method (GB/T 1548–2004), and cuprammonium hydroxide (Cuoxam) method (GB/T 9170–1999).

In this work, the cellulose samples were dissolved in TBAH/DMSO/H_2_O (THDS, with the ratio of 1:8:1 in weight) to generate true molecular solutions with five gradient concentrations. The relationship between the DP or molecular weight and the intrinsic viscosity of cellulose in THDS was established in the form of a Mark–Houwink equation, according to the DP value tested by Cuoxam method. Then, the reliability of the equation was checked by CED method, as well as the gel permeation chromatography (GPC) method to test the corresponding cellulose acetate samples. Furthermore, the morphology of cellulose molecules in THDS solvent system was characterized by atomic force microscopy (AFM), and the dispersion level of cellulose molecules in THDS solution was analyzed by dynamic light scattering (DLS). The stability of cellulose molecules in the THDS solution was studied by comparing the DP values of the equivalent regenerated samples with the original cellulose. The crystal structure of the cellulose samples were tested by wide-angle X-ray diffraction (WAXD), Fourier transform infrared (FT-IR), and ^13^C NMR, before and after regeneration from the THDS solutions.

## 2. Materials and Methods

### 2.1. Materials

Materials and reagents: Straw cellulose (Isolated in our lab with the reported method [44]), wood pulp (Zhonglin Industry Co., Ltd., Canada), cotton pulp (Xinxiang Chemical Fiber Co., Ltd., Xinxiang, Henan, China), bamboo pulp (Chengdu Liya Co., Ltd., Chengdu, Sichuan, China), and microcrystalline cellulose (MCC, Shanghai Aladdin Biochemical Technology Co., Ltd., Shanghai, China) were adopted as the cellulose samples for testing. The samples were codes as C1–C5 sequentially, with the increase of their molecular weights. Other two kinds of bamboo pulp with different degrees of polymerization (Chengdu Liya Co., Ltd., Chengdu, Sichuan, China) were codes as C6–C7 sequentially. Samples C1–C5 were used for equation derivation and samples C6–C7 for equation verification. Cellulose samples were all oven dried at 105 °C for 2 h before use. Tetra-n-butylammonium hydroxide (TBAH, 15 wt %) was purchased from Alfa Aesar Co. The TBAH aqueous solution was concentrated to 50 wt % under vacuum before use. Dimethyl sulfoxide (DMSO, A.R. grade) and acetylation reagent acetic anhydride (Ac_2_O, A.R. grade) were supplied by Chengdu Haihong Experimental Instrument Co., Ltd., Chengdu, Sichuan, China. The Cuoxam and the CED solution were prepared according the standard methods.

### 2.2. Preparation of Dilute Cellulose Solution

#### 2.2.1. Preparation of Cellulose/Cuoxam Solution

The Cuoxam solution was prepared by dissolving copper carbonate in ammonium hydroxide at room temperature under continuous stirring, according to the standard method (GB/T 9170–1999). The content of copper in Cuoxam solution should be greater than 1.3 g/100 mL. The cellulose samples C1–C7 were separately dissolved in Cuoxam solution under room temperature with magnetic stirring to give a blue homogeneous solution with the concentration of about 1.5 g/L, to test the degree of polymerization (DP_C_).

#### 2.2.2. Preparation of Cellulose/CED Solution

The CED solvent was prepared by using newly synthesized copper hydroxide and ethylenediamine according to the standard method (GB/T 1548–2004). The copper content in the CED solution was determined to be 1.00 ± 0.02 mol/L by titration method. The cellulose samples C1–C5 were completely dissolved in CED by continuous vigorous magnetic stirring at room temperature for 7 days. The concentration of cellulose/CED solution was controlled between 3 and 4 g/L to test the degree of polymerization (DP_CED_).

#### 2.2.3. Preparation of Cellulose/THDS Solution

The DMSO was evenly mixed with the cellulose samples C1–C5 at room temperature. Then, the TBAH aqueous solution was added under mild stirring for 12 min to obtain the transparent cellulose solution with the concentration of 0.25 wt %, which was taken as the mother liquor. A series of the diluted cellulose solutions were prepared by gradient dilution, as shown in Table 1.

### 2.3. Viscosity Analysis of the Cellulose Solution by Ubbelohde Viscometer

#### 2.3.1. Cellulose/Cuoxam Solution

The flow time of Cuoxam (*t*_0_, s) and the corresponding solution (*t*_1_, s) were tested with the Ubbelohde viscometer under 25 °C. The intrinsic viscosity ([η]) was determined via Huggins’ plot (i.e., the plot of η_sp_/*c* versus *c*) based on Equation (1), where c was the cellulose concentration in unit of g/L, and *K*_1_ is a constant. The specific viscosity (η_sp_) was calculated based on Equation (2). The intrinsic viscosity [η] values of the cellulose samples C1–C7 in Cuoxam were recorded for further calculating the DPc. (1)$$\eta_{\mathrm{sp}}/c=\left[\eta \right]+K_{1}\;{\left[\eta \right]}^{2}c.$$
(2)$$\eta_{\mathrm{sp}}=t_{1}/t_{0}-1.$$

#### 2.3.2. Cellulose/CED Solution

The outflow time (*t*) of the cellulose/CED solutions of samples C1–C5 were tested with the Ubbelohde viscometer under 25 °C, in the unit of s. According to the one-to-one correspondence between the viscosity ratio ([η]/η) and [η]·*c*. (GB/T 1548–2004), and Equation (3), the intrinsic viscosity [η] of the cellulose samples C1–C5 in CED solution would be calculated.
(3)[η]η=h·t.
Here, *h* is the constant of viscosity meter in unit of s^−1^. The constant of viscosity meter (*h*) was determined according to the standard method by glycerin aqueous solution calibration.

The intrinsic viscosity [η] of the cellulose samples C1–C5 in CED solution would be further adopted to calculate the DP values measured by CED method (DP_CED_), and comparably analyzed with the Cuoxam and THDS solutions.

#### 2.3.3. Cellulose/THDS Solution

The capillary Ubbelohde viscometer with inside diameter of 0.8–0.9 mm was used to measure the flow time (*t*_0_ and *t*) of the cellulose solution and the solvent. The water bath temperature was set at 25 °C. All the tests were conducted for more than 5 times. Only the values that met the requirement of Grubbs’ Test (α = 0.01), Dixon’s Test (α = 0.01), and Student’s t test (*t* = 5.598) were selected for elucidating the intrinsic viscosity-molecular weight relationship for cellulose in THDS solution.

### 2.4. Determination of the Degree of Polymerization (DP) of Cellulose

#### 2.4.1. Cuoxam Method

The degree of polymerization of the cellulose samples C1–C7 in Cuoxam solutions were analyzed according to Equation (4), and noted as DP_c_.
(4)$$\mathrm{DP}=\frac{\eta_{\mathrm{sp}}}{\left(1+0.29\eta_{\mathrm{sp}}\right)\times 5\times 10^{-4}\times c}$$
Here, η_sp_ is the specific viscosity of the Cuoxam solution, and *c* is the concentration of cellulose, in the unit of g/L.

The DP_C_ values of the cellulose samples C1–C5 were implicated to determine the Mark–Houwink equation for cellulose in THDS solution. The DP_C_ values of samples C6 and C7 were used for the difference analysis between Cuoxam method and THDS method.

#### 2.4.2. CED Method

According to the intrinsic viscosity of the cellulose/CED solution obtained above, the DP_CED_ of the cellulose samples were calculated according to Equation (5). (5)$$\mathrm{DP}={\left(\frac{\left[\eta \right]}{2.28}\right)}^{1.316}.$$
Here the unit of the intrinsic viscosity [η] is mL/g.

The DP_CED_ values of samples C1–C5 were applied for analyzing the difference of the DP values of the cellulose samples obtained by different methods.

#### 2.4.3. THDS Method

The following Equation (7) was adopted for THDS method. (6)$$\left[\eta \right]=K_{2}\times {\mathrm{DP}}^{\alpha }.$$
Here [η] is the intrinsic viscosity of cellulose/THDS solution, in the unit of mL/g, *K*_2_ and *α* are parameters which depend on the solvents. DP is the degree of polymerization of the cellulose sample, which is tested by the Cuoxam method. The intrinsic viscosity [η] is determined by using an Ubbelohde viscometer via Huggins’ plot (i.e., plot of η_sp_/*c* versus *c*) according to Equation (1).

#### 2.4.4. Determination of the DP of the Derived Cellulose Acetate Samples by the GPC Method

The cellulose samples C1–C5 were transferred into the corresponding acetylated cellulose samples with acetylation reagent Ac_2_O in the THDS solvent. According to the reported procedures [45,46,47,48,49]. At first, 1 wt % cellulose was dissolved in THDS, then, 13 mol Ac_2_O per mole of AGU was added into the solution. The reaction mixture was stirred at 60 °C for 3 h. The cellulose acetate samples (CA) were precipitated by pouring the reaction mixture into 500 ml ethanol under continuous stirring. After filtration, the crude cellulose acetate samples were washed three times with ethanol, and then vacuum dried at 60 °C. The prepared cellulose acetate samples were named as CA1–CA5, corresponding to the five cellulose samples C1–C5. FT-IR spectra proved the successful acetylation of cellulose. The degrees of substitution (DS) were calculated by ^1^H NMR spectra method (Appendix A).

A gel permeation chromatography (HLC-8320GPC, Tosoh Corporation, Shanghai, China) instrument with two columns in series (TSK gel Super AWM-H, Tosoh Corporation, Shanghai, China) connected to refractive index detector, was used to calibrate the molecular weight of the cellulose acetate samples. The GPC test was performed with N, N-dimethyl formamide (DMF) as eluent, polymethyl methacrylate as a standard at 40 °C. The flow rate of eluent was set as 0.4 mL/min. The time was 30 min for one sample.

The DP values measured by GPC method (DP_GPC_) of the cellulose samples (C1–C5) were calculated by Equation (7), based on the molecular weight of cellulose acetate samples (CA1–CA5). Where, *M*_w_ is the weight-average molecular weight of the cellulose acetate samples, DS is the degree of substitution of acetylated cellulose. The DP value of the cellulose sample was calculated by dividing the molecular weight of the corresponding cellulose acetate by the weight of the repeating units [9].
(7)$$\mathrm{DP}=\frac{M_{w}}{43\times \mathrm{DS}+162}.$$

### 2.5. The Analysis of Cellulose Molecules Dispersion Level in THDS Solution

#### 2.5.1. Dynamic Light Scattering (DLS)

The cellulose solution (C1, 0.5 × 10^−4^ g/mL) was prepared in the similar way as shown in Section 2.2.3. The prepared solution was filtrated through a 0.25 μm PTFE Millipore filters into a clean vial. The dispersion level of the cellulose molecules in the THDS solution was tested by dynamic light scattering (DLS, wide-angle laser scattering instrument (Nano Brook Zeta PALS; Brookhaven Instruments, Holtsville, NY, USA)). The refractive index of the cellulose/THDS solution was 1.466. DLS data was analyzed using the CONTIN program. Meanwhile, the Stokes–Einstein relationship was used to calculate the hydrodynamic radius (*R*_h_). (8)Rh=kBT6πη0D,
where *k*_B_ is the Boltzmann constant; *T* is the temperature, in the units of K; η_0_ is the solvent viscosity, in the unit of cP; and *D* is the translational diffusion coefficient.

#### 2.5.2. Atomic Force Microscope (AFM)

Multimode scanning probe microscope (AFM, Veeco, Santa Barbara, CA, USA) operated in the tapping mode was used for the imaging of the 0.05 mg/mL cellulose solution sample (C5). The morphological data of the images were analyzed using the AFM-accessory software.

### 2.6. The Structure Analysis of Cellulose in THDS Solution

The cellulose/THDS solution was added into excessive deionized (DI) water under continuous stirring to completely regenerate the cellulose, which was named RC. The RC was washed with DI water, and filtered through a 0.22 μm filter until the filtrate was neutral. The washed RC was freeze-dried for further use and its DP value was tested bythe THDS solution method to compare with original cellulose sample.

Fourier transform infrared (FT-IR) spectra were recorded on a Nicolet 6700 (Thermo Scientific Inc., Waltham, MA, USA) in room temperature, using the KBr tablet method. The spectral resolution was 2 cm^−1^. In the sample preparation and measurement process, infrared light was used to dry to remove moisture.

The crystal structures of the original cellulose (C6 and C7) and regenerated samples (RC6 and RC7) were characterized by wide-angle X-ray diffraction (WAXD, Panalytical X’pert PRO diffractometer, Ni-filtered Cu *K*α radiation, Wageningen, The Netherlands), at 40 kV and 30 mA. The scanning angle ranged from 5° to 50°.

The aggregation state structure of C6, C7, RC6, and RC7 was analyzed by solid-state ^13^C nuclear magnetic resonance (NMR). The solid-state ^13^C NMR spectra were recorded on a AVANCE III 400 apparatus (Bruker Biospin GmbH, Rheinstetten, Germany) with an operating frequency of 75.5 MHz, a magic angle spinning (MAS) rate of 14 kHz, compensation time of 34 ms, contact time of 2 ms, and delay of 2 s between the two pulses.

## 3. Results and Discussions

### 3.1. Preparation and Viscosity Analysis of the Cellulose in THDS Solution

All cellulose samples C1–C9 were dissolved in THDS at room temperature within 12 min to obtain the transparent solutions as shown in Appendix A. A quick dissolving process for cellulose in THDS solvent was provided as Appendix A.

The plots of the reduced specific viscosity η_sp_/*c* were drawn versus the concentration c, as shown in Figure 1. The line was extrapolated to zero concentration to obtain the intrinsic viscosity ([η]) of the cellulose solution in THDS. The lopes of the lines are corresponded to the Huggins’ constant *K*_1_ values. The values of [η] and the corresponding *K*_1_ in the Equation (1) of the cellulose samples were summarized in Table 2.

The intrinsic viscosity of each specific sample in three solvents are listed in Table 3. They are in the sequence of [η]_Cuoxam_ < [η]_CED_ < [η]_THDS_. According to the hydrodynamic theory of polymer in dilute solution, the intrinsic viscosity of a polymer is proportional to the hydrodynamic volume (*V*_e_/*M*) of a polymer per unit mass in solution. The value of [η] increased with the interaction between the solvent and the polymer molecules. So, the solvation effect of THDS is much better than both the solvation effect of the Cuoxam and the CED. The improvement of [η] in all the three solvents with the order of cellulose samples C1–C5 are ascribed to the increasing of the DP.

### 3.2. The Elucidation of the Mark–Houwink Equation for Cellulose/THDS Solution

The DP values of the cellulose samples C1–C5, measured by the Cuoxam method (as shown in Table 4), were used to determine the Mark–Houwink equation for the cellulose/THDS dilute solution. The intrinsic viscosity values of the cellulose samples C1–C5 in the THDS solvent were correlated to the DP_c_ values, as shown in Figure 2. The power law was founded from the ln–ln plot, with the *α* value as 1.21, and the *K*_2_ value as 0.24. (9)$$\left[\eta \right]=0.24\times {\mathrm{DP}}^{1.21}.$$
Or (10)$$\left[\eta \right]=5.09\times 10^{-4}M_{\eta }{}^{1.21}.$$
Here *M*_η_ is the viscosity average molecular weight by dilute solution viscosity method of the cellulose samples.

The correlation coefficient (Pearson’s r) was 0.981 by linear regression analysis, which represented a good reliability of the Mark–Houwink equation. Where, the *α* value was 1.21, which indicated that the linear flexible chains of cellulose were dispersed in the THDS solvent.

The Mark–Houwink equation for cellulose in PF/DMSO at 30 °C is reported as $\left[\eta \right]=3.01\times {\mathrm{DP}}^{0.81}$ [12]. In 1-butyl-3-methylimidazolium acetate/dimethyl sulfoxide (BmimAc/DMSO) solution, the relationship between intrinsic viscosity and DP is proved to be $\left[\eta \right]=2.5\times {\mathrm{DP}}^{0.83}$ [9]. The reported values of the α in the Mark–Houwink equation were summarized in Appendix A, including Cadoxen [9,12,50], FeTNa [51], PF/DMSO [12], Cuen [12,50], Cuoxam [12,50], BmimAc/DMSO [9], NaOH/urea aqueous solution [30,32,33], LiOH aqueous solution [27,28,52], NH_3_/NH_4_SCN [50], CED, and DMAc/LiCl [25,26]. The *α* in the Mark–Houwink equation indicates the solubility of the solvent for cellulose. The α values of 0.8 and above means linear flexibility of the macromolecular chains in the solvent [53].

The dissolution state of cellulose in THDS solvent was evaluated by DLS characterization (Figure 3), showed a single distribution, and the average particle size of cellulose was around 7 nm. In brief, cellulose is dispersed as isolated molecules in THDS solution, which consisted of the previous report on static light scattering and small-angle neutron scattering tests [13].

In the two-dimensional (2D) AFM height image (Figure 3) of sample C5, we observed distinct randomly-distributed lines of slender roots on the surface of the amino silicon wafer. Since cellulose is the only solute in the preparation process, these lines certainly belong to cellulose [54]. Moreover, we considered this two-dimensional structure of a single cellulose chain, because cross-sectional analysis showed that the height of this two-dimensional structure was 0.3 to 1 nm, consistent with the diameter of a single cellulose chain [55]. This conclusion showed that the THDS solvent was a suitable solvent for cellulose.

### 3.3. Comparable Analysis of Cellulose DP

To verify the reliability of the Mark–Houwink equation for the THDS system, the corresponding cellulose samples (C1–C5) were measured by the CED method, and the DP values were collected in Table 4. The DP values determined by the Cuoxam method were lower than those obtained by the CED method. The difference between the two methods was about 21% to 36%. Similar results are reported, and the DP values of cellulose tested by the CED method were higher than those obtained by the Cuoxam method [50]. Moreover, cellulose was partially degraded in both the Cuoxam and the CED solutions.

The cellulose samples C1–C5 were derived into the cellulose acetate samples CA1–CA5 for GPC analysis. The derivative samples were all structurally manifested by FT-IR. The DS values were determined by ^1^H NMR, according to the literature [10,11,56,57]. The DS values of the CA1–CA5 were 1.03, 0.87, 1.49, 0.97, and 0.89, respectively. The FT-IR spectra and the ^1^H NMR results were provided in the Appendix A.

The GPC curves of the derived cellulose samples CA1–CA5 are shown in Figure 4. Table 5 shows the average molecular weight characterized by GPC method, including weight-average molecular weight (*M*_w_), viscosity-average molecular weight (*M*_η_), number-average molecular weight (*M*_n_), z-average molecular weight (*M*_z_), peak-molecular weight (*M*_p_) of the derived cellulose, and the corresponding molecular weight distribution coefficients (MWD, *M*_w_/*M*_n_). In addition, the *M*_w_, *M*_η_, *M*_n_ of cellulose were calculated from *M*_w_. Generally, the molecular weight of the polymer is in the order of *M*_w_ > *M*_η_ > *M*_n_. The molecular weight distribution is the relationship between the relative content of each molecular weight close fractions in the homopolymer homologue and the molecular weight. The *M*_w_/*M*_n_ value of cellulose ranged from 3 to 7, which indicated the increasing polydispersity of the cellulose acetate samples CA1 to CA5.

As shown in Table 4, the DPs calculated by *M*_w_ of GPC results were considerably higher than those tested by the Cuoxam method. The general order of *M*_w_ > *M*_η_ > *M*_n_ and the cellulose degradation in the copper ethylenediamine solution lead to this result [58]. As shown in Table 4, the DP of C6 and C7 determined by the Cuoxam method were about 21–22% lower than those specified by the THDS method. Because the polydispersity is ignored in the Mark–Houwink equation [50], the differences in the viscosity-based DP values are considered to be within a reasonable error range.

### 3.4. The Structure Analysis of Cellulose in THDS

According to the FT-IR spectra (Figure 5), the peak at 2898 cm^−1^ was ascribed to the stretching vibration of C–H. The peaks at 1650 and 1370 cm^−1^ were attributed to the adsorbed water and the deformation vibration of C–H, respectively; two peaks at 1158 and 896 cm^−1^ were asymmetric stretching vibration and out-of-plane deformation vibration of β-glycosidic bond C–O–C, respectively. C–O stretching vibration peak around 1114 cm^−1^ was split into several peaks. It is worth noting that early studies indicate that the peak at 1418 cm^−1^ is the characteristic peak for cellulose II and amorphous cellulose [59]. In contrast, if the sample is rich in cellulose I, characteristic absorption peaks appear at 1430 and 1111 cm^−1^ [60]. There was apparent blue shift happened to the O–H stretching vibration when the cellulose samples C6 and C7 regenerated from their THDS solutions, which were named as RC6 and RC7, seperately. The peaks shifted from 3433 to 3444 cm^−1^, and 3342 to 3389 cm^−1^, corrrespondingly, because of the transformation of the crystal structure from cellulose I to cellulose II. No new covalent bond formed in the regenerated cellulose.

According to the WAXD analysis (Figure 5), the cellulose samples (C6 and C7) showed typical diffraction peaks of cellulose I at 14.9°, 16.5°, 22.7°, and 34.6°, respectively, corresponding to the 1$\bar{1}$0, 110, 200, and 004 crystal surfaces. However, the regenerated cellulose (RC6 and RC7) showed three distinct diffraction peaks at 12.3°, 20.0°, and 22.1°, which were typical diffraction peaks of cellulose II crystal form, corresponding to the 1$\bar{1}$0, 110, and 020 crystal faces, respectively. As was shown in Figure 5, the crystallinity of the regenerated cellulose samples (210.6‰ for RC6 and 174.5‰ for RC7) apparently decreased compared with the original cellulose samples (259.6‰ for C6 and 278.6‰ for C7), which was mainly because of the quick regeneration process by adding excessive water into the cellulose solution.

The same results were shown in the solid-state ^13^C NMR spectra (Figure 5). The signal at 90–95 ppm corresponded to the highly ordered crystallized cellulose, and the signal at 83–90 ppm corresponded to the disordered and the less-ordered cellulose chains [61,62,63,64,65,66,67]. In addition, various crystal types of cellulose can be identified from the splitting of the peaks [64]. As can be seen in Figure 5, the splitting of the C4 signal was the most prominent. However, the peak separation between amorphous and crystalline was not apparent after cellulose regeneration. It indicated that the degree of crystal order increased after cellulose regeneration. As could be seen from Figure 5, the crystalline region of regenerated cellulose was mostly composed of cellulose II crystal form, while the original cellulose was cellulose I crystal form. So, the variance in the [η]s and DPs was most likely due to the recrystallization of the original cellulose I into cellulose II after regeneration from the solution [43,68]. The cellulose II is less soluble than the original cellulose I [68], leading to a much smaller intrinsic viscosity and a corresponding lower calculated DP value in the solution of the regenerated cellulose.

Then the DP values of cellulose samples regenerated from the corresponding THDS solutions were investigated, as shown in Table 6, and [η]s and DPs measured by the THDS method of the original (C6 and C7) and the regenerated cellulose (RC6 and RC7) in their THDS solutions. There were 16% and 14% variance in the [η]s (([η]_Cn_ − [η]_RCn_)/[η]_Cn_) and variance in the DPs ((DP_Cn_ − DP_RCn_)/DP_Cn_) between the original and the regenerated cellulose, respectively.

## 4. Conclusions

This work reports a convenient and reliable method for determining the molecular weight of cellulose via the intrinsic viscosity, and discloses the cellulose dispersed in THDS at a molecular level by AFM and DLS characterization. The relationship between the intrinsic viscosity and DP of cellulose was determined as $\left[\eta \right]=0.24\times {\mathrm{DP}}^{1.21}$ in the dilute THDS solution. The α value of 1.21 in the Mark–Houwink equation also proved the molecular chains stretched well in the THDS solvent. The reliability of the equation was verified by comparing the cellulose DP values tested by this method with those obtained by the Cuoxam method, the CED method, and the GPC method, separately. What is more, the structure analysis proved that cellulose was dissolved in the THDS solvent at the molecular level without derivatization, and was transferred into cellulose II from original cellulose I after regeneration. Furthermore, we found that the TBAH/DMSO aqueous solution provided a beneficial medium for the homogeneous acetylation of cellulose. The controllable acetylation of cellulose in such medium is still under study.

## Figures and Tables

**Figure 1 polymers-11-01605-f001:**
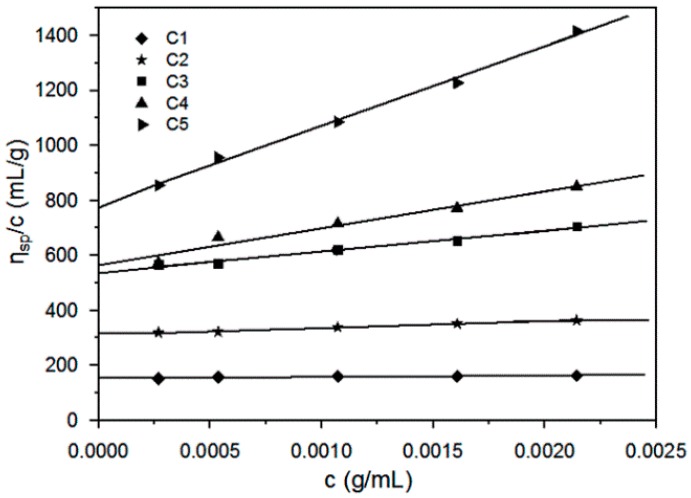
Condensed viscosity (η_sp_/*c*) and concentration of five cellulose samples in the THDS.

**Figure 2 polymers-11-01605-f002:**
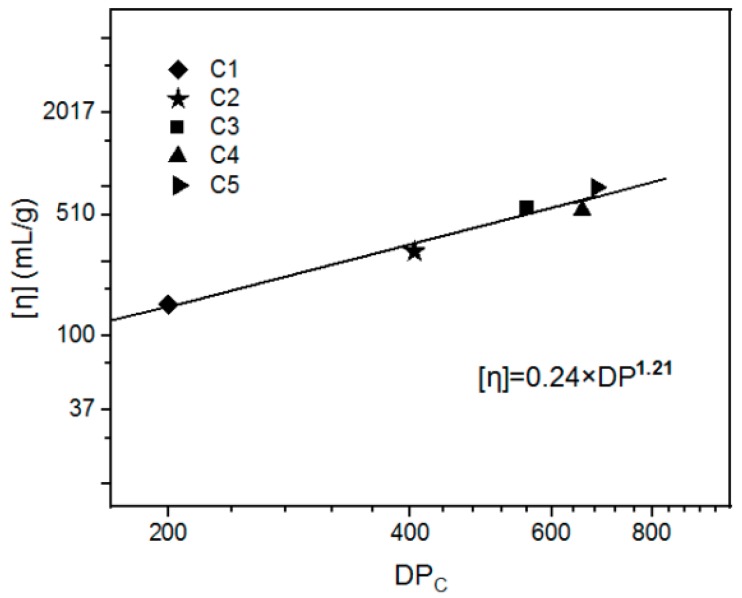
The relationship between the [η] and the DP_C_ of cellulose.

**Figure 3 polymers-11-01605-f003:**
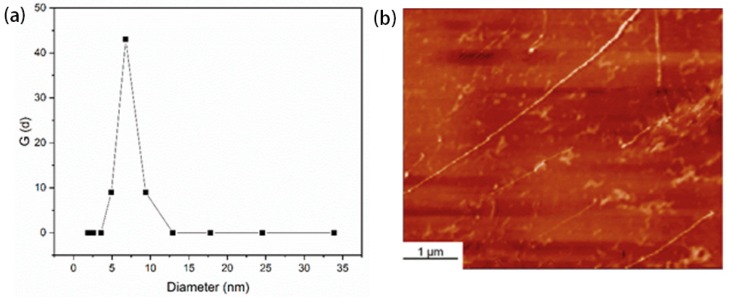
The dissolved state of cellulose in THDS solution: (**a**) DLS profiles of the cellulose (C1) solution in THDS with concentration of 0.5 × 10^−4^ g/mL, at 25 °C; (**b**) AFM image of the dilute cellulose solution in THDS observed on an amino silicon wafer adsorption.

**Figure 4 polymers-11-01605-f004:**
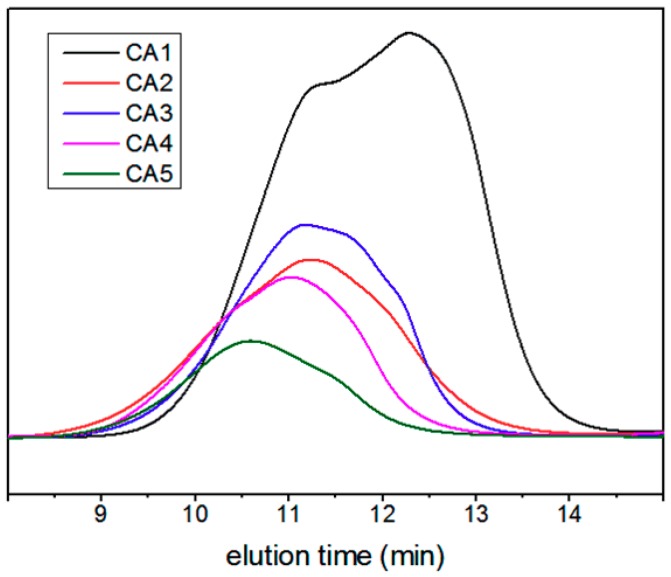
The GPC curves of the derived cellulose samples of CA1–CA5.

**Figure 5 polymers-11-01605-f005:**
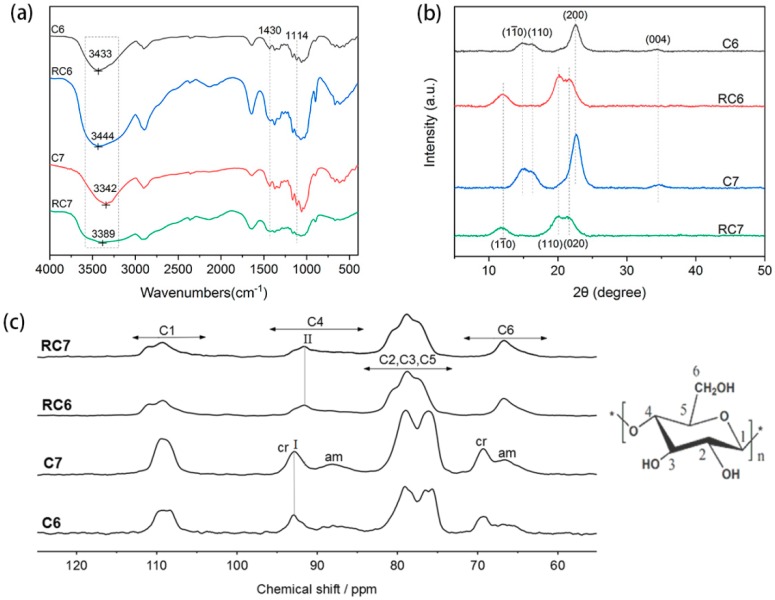
The structure analysis of cellulose samples (cellulose before and after regeneration) in THDS: (**a**) The FT-IR spectra of cellulose samples; (**b**) the XRD spectra of cellulose samples; (**c**) the solid-state ^13^C NMR spectra of cellulose samples: cr corresponding to the crystalline signal; am corresponding to amorphous signal; I corresponding to cellulose I crystal form; II corresponding to cellulose II crystal form.

**Table 1 polymers-11-01605-t001:** Preparation of cellulose/THDS aqueous solutions by dilution method.

Concentration (wt %)	Mother Liquor (mL)	TBAH (50% *w*/*w*, mL)	DMSO (mL)
0.05	6	5.34	18.66
0.10	12	4.00	14.00
0.15	18	2.67	9.33
0.20	24	1.34	4.66

**Table 2 polymers-11-01605-t002:** The intrinsic viscosity ([η]) and the *K*_1_ values of cellulose solution in THDS at 25 °C.

Parameters	Samples						
	C1	C2	C3	C4	C5	C6	C7
[η] (mL/g)	152.8	310.6	538.8	564.5	738.6	550.6	693.4
*K* _1_	0.2	0.3	0.3	0.4	0.5	0.2	0.4

**Table 3 polymers-11-01605-t003:** Comparison of the intrinsic viscosity the cellulose solutions.

Intrinsic Viscosity (mL/g) *	Sample				
	C1	C2	C3	C4	C5
[η]_Cuoxam_	29.5	68.1	137.2	277.4	279.9
[η]_CED_	-	281.6	335.8	419.9	458.1
[η]_THDS_	152.8	310.6	538.8	564.5	738.6

* The intrinsic viscosities of the samples in the three solvents are presented as [η]_Cuoxam_, [η]_CED_, and [η]_THDS_, respectively.

**Table 4 polymers-11-01605-t004:** The DP values of cellulose samples C1–C7 determined by different methods.

DP	Sample						
	C1	C2	C3	C4	C5	C6	C7
DP_C_	200	405	557	655	682	486	600
DP_CED_	275	563	713	958	1073	-	-
DP_GPC_ ^a^	232	556	731	983	-	-	-
DP_THDS_	-	-	-	-	-	616	752
(DP_CED_-DP_C_)/DP_CED_ (%)	27	28	21	31	36	-	-
(DP_GPC_-DP_C_)/DP_GPC_ (%)	13	27	23	33	-	-	-
(DP_THDS_-DP_C_)/DP_THDS_ (%)	-	-	-	-	-	21	22

^a^ The DP_GPC_ values were calculated from the corresponding *M*_w_.

**Table 5 polymers-11-01605-t005:** The molecular weight and the distribution of cellulose acetate samples of CA1–CA5 tested by the GPC method.

Samples	Molecular Weight					
	*M* _n_	*M* _w_	*M* _z_	*M* _η_	*M* _p_	*M*_w_/*M*_n_
CA1	11784	47898	252932	47898	15527	4.1
CA2	32411	111016	1039096	111016	45753	3.4
CA3	49199	165543	980068	165543	57839	3.4
CA4	29147	200420	4247791	200420	44473	4.9
CA5	75664	373374	4805672	373374	98679	6.9

**Table 6 polymers-11-01605-t006:** Comparison of the intrinsic viscosity and DP between the original cellulose solution and cellulose solution after regeneration.

Sample	Parameter			
	[η] (mL/g)	([η]*_C_*_n_ − [η]*_RC_*_n_)/[η]*_C_*_n_ (%)	DP	(DP*_C_*_n_ − DP*_RC_*_n_)/DP*_C_*_n_ (%)
C6	550.6	16	616	14
RC6	459.8		528	
C7	693.4	16	752	14
RC7	579.9		644	

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
