# Peer review of "Elucidation of the Relationship between Intrinsic Viscosity and Molecular Weight of Cellulose Dissolved in Tetra-N-Butyl Ammonium Hydroxide/Dimethyl Sulfoxide"

_polymers, 2019, doi:10.3390/polym11101605_

Round 1

Reviewer 1 Report

Manuscript ID polymers-586511

The subject addressed in the present manuscript is interesting. However, the present work lacks of some novelty. A work presented by Liu et al. (Liu, J.; Zhang, J.; Zhang, B.; Zhang, X.; Xu, L.; Zhang, J.; He, J.; Liu, C.-Y. Determination of intrinsic viscosity-molecular weight relationship for cellulose in BmimAc/DMSO solutions. Cellulose 2016, 23, 2341-2348, doi:10.1007/s10570-016-0967-1) reports a similar work using a different solvent system (BmimAc/DMSO). On the other hand, some works were already presented reporting the good performance of TBAH based systems to dissolve cellulose into a molecular level (Chen, X.; Chen, X.; Cai, X.-M.; Huang, S.; Wang, F. Cellulose Dissolution in a Mixed Solvent of Tetra(n-butyl)ammonium Hydroxide/Dimethyl Sulfoxide via Radical Reactions. ACS Sustainable Chemistry & Engineering 2018, 6, 2898-2904, doi:10.1021/acssuschemeng.7b04491. or Alves, L.; Medronho, B.; Antunes, F.E.; Topgaard, D.; Lindman, B. Dissolution state of cellulose in aqueous systems. 1. Alkaline solvents. Cellulose 2016, 23, 247-258, doi:10.1007/s10570-015-0809-6. or Gubitosi, M.; Duarte, H.; Gentile, L.; Olsson, U.; Medronho, B. On cellulose dissolution and aggregation in aqueous tetrabutylammonium hydroxide. Biomacromolecules 2016, 17, 2873-2881, doi:10.1021/acs.biomac.6b00696.). Thus authors need to highlight the main achivements that lead to an advance in state of art, obtained during the present work. The conclusions stated in the current manuscript present some similarity with those presented in the work of Liu et al.

In addition, the english language needs a deep improvement, mainly in introduction section. Several sentences make no sense, examples: Page 1, line 39-41: “Because cellulose can hardly be dissolved in common solvents, due to its extensive intra- and intermolecular hydrogen bonding [9–12], and the amphiphilicity distribution of the crystal surfaces [13–15].”, something is missing in the sentence; Page 2, line 63: “…generate genuine dilute solutions…”, genuine is not a scientific term, at least in the present context. Page 3, line 93: “The content of copper and the ammonia content ammonia solution to be greater than...”, very confusing phrase. Page 3, line 101: “…by continues vigorous…”, maybe continuous instead of “continues”; Page 3, line 107: “A serious of the diluted cellulose…”, maybe the authors want to say “a series of diluted…”. Page 6, line 210: “In the sample preparation and measurement process, used infrared light to bake to remove moisture.” The sentence needs to be improved and bake is not a scientific term. Please become the manuscript text clear and scintifically correct.

Also the chainning of the ideas needs to be improved. Example: Page 1, line 37-42: “The accurate determination of molecular weight is one of the key issues for the development of cellulose based materials in laboratory research, as well as industry. Because cellulose can hardly be dissolved in common solvents, due to its extensive intra- and intermolecular hydrogen bonding [9–12], and the amphiphilicity distribution

of the crystal surfaces [13–15]. The traditional solvent, C2S/NaOH used in viscose fiber production with sulfuric acid as coagulation bath, causes serious pollution to the air and water.” Authors mixed three ideas without any close relationship. Please become the text more fluid and easy yo follow.

During WAXD results discussion athors stated “As could be seen from the Figure 5, the crystalline region of regenerated cellulose was mostly composed of cellulose Ⅱ crystal form, while the regenerated cellulose was cellulose I crystal form.”. The discussion is again very confusing and wrong. Once dissolved and regenerated cellulose turns into cellulose II. Is not possible obtain cellulose I from a dissolved cellulose solution.

Page 2, line 48: Authors used the reference “Alves, L.; Medronho, B.; Antunes, F.E.; Topgaard, D.; Lindman, B. Dissolution state of cellulose in aqueous systems. 2. Acidic solvents. Carbohydr. Polym. 2016, 23, 247–258.” to refer the dissolution of cellulose in alkaline solvent systems. However this reference deals with cellulose dissoltion in acidic systems. The same authors have some works dealing with alkaline solvent systems (Alves, L.; Medronho, B.; Antunes, F.E.; Topgaard, D.; Lindman, B. Dissolution state of cellulose in aqueous systems. 1. Alkaline solvents. Cellulose 2016, 23, 247-258, doi:10.1007/s10570-015-0809-6. or Alves, L.; Medronho, B.F.; Antunes, F.E.; Romano, A.; Miguel, M.G.; Lindman, B. On the role of hydrophobic interactions in cellulose dissolution and regeneration: Colloidal aggregates and molecular solutions. Colloids and Surfaces A: Physicochemical and Engineering Aspects 2015, 483, 257-263, doi:http://dx.doi.org/10.1016/j.colsurfa.2015.03.011. or Alves, L.; Medronho, B.; Filipe, A.; E. Antunes, F.; Lindman, B.; Topgaard, D.; Davidovich, I.; Talmon, Y. New Insights on the Role of Urea on the Dissolution and Thermally-Induced Gelation of Cellulose in Aqueous Alkali. Gels 2018, 4, 87 or Gustavsson, S.; Alves, L.; Lindman, B.; Topgaard, D. Polarization transfer solid-state NMR: a new method for studying cellulose dissolution. RSC Advances 2014, 4, 31836-31839, doi:10.1039/C4RA04415K). Please correct the references in text.

Some tecnhiques are wrongly discribed. For example, in the section 2.6, line 206 of page 6, authors refer the DP determination inside the FTIR description: “And the DP of the regenerated cellulose was obtained by THDS method.” Also SLS is referred in conclusion section. They have used DLS and not SLS.

Reviewer 2 Report

Hi

The article was written in a good shape and discussed properly, However, I am going to mention following comments about that

Different cellulose samples were used to evaluate correlation between intrinsic viscosity and molecular weight, it will be valid if only we have pure cellulose. The authors didn't mentioned purity of samples. Based of the XRD results of sample C6 and C7 which have 25-28% crystallinity, it is appear that the samples are not pure cellulose. presence of other carbohydrate in the samples like hemicellulose can effect of its solubility, viscosity and the results are not representative for cellulose anymore Correlation between viscosity and molecular weight were reported during last century in tons of articles and books and are not universal standard methods due to effect of old cellulose solvents on its degradation. based on XRD figure of samples, it seems the crystallinity of the C6 and specially C7 is much more than reported numbers.   It is recommend to use non-degradable cellulose solvent like ionic liquid for the study instead of old solvents

Reviewer 3 Report

The present contribution gives some new insight into how to determine the molecular weight of cellulose and should be published after major revision in the journal “Polymers”.

The abstract must be self-explanatory without reading the full text. Define “alpha” (line 19) and “DP” (line 24) where first used. Font size: “according” (line 65), “Then, the reliability” (line 66), “molecules” (line 72), “[39]” (line 76), [40-44] (line 165), [45] (line 183, 274 and 276), “[23]” (line 273 and 276), “[23,45,46” and “[47]” (line 275), “[23,46]” (line 276, twice), “[26,28,48]” and [21,22,49] and “[46]” (line 277), “[16,179” (line 278), “[50]” (line 280), “[51]” (line 280), [52]” (line 292), “[53]” (line 295), “[46]” (line 303), “[54-57]” (line 307), “[58]” (line 326), “[46]” (line 328), “[59]” (line 337), “[60]” (line 338), “[61-67]” (line 359), “[64]” (line 361), “[38,68]” and “[68]” (line 367). Use upper and lower case correctly: “Copper” (line 59), “Cuprammonium” (line 60), “Microcrystalline” (line 79), “china” (line 80), “Degree” (line 171), “the Netherlands” (line 215), “equation” (line 234). “-0.098 MPa” (line 84/85) – What does the negative value of pressure mean? “0.0015 g/mL” (line 96) – It may be more comprehensible for the readers to be replaced with 1.5 g/L. The same is with “0.003 g/mL” and “0.004 g/mL” (line 102/103). Define symbols and abbreviations where first used in the text: “DPc” (line 116), “K1” and “C” (line 118), “[eta]/eta” (line 123), “DPCED” (line 130), “DMF” (line 176), “DPGPC” (line 178), “M” (line 260), „BmimAc/DMSO” (line 273), “Mw/Mn” (line 312), “Mw” and “Mn” and “Meta” (line 313), “Mz” and “Mv” and “Mp” and “Mz/Mw” (Table 5), “[eta]Cn” and “[eta]RCn” and “CPCn” and “CPRCn” (Table 6). “G (alpha=0.01), Q (alpha=0.01), and t (t=5.598)” – What does it mean? Equation (4) (line 142) – Clarify denominator and numerator. Uniform use of upper and lower case in all headings: line 162, 185, 186, 198 and 227. “Solid-state 13C NMR Solid” – Check if correct. Table 2, Table 3, Table 5 and Table 6: Decimal places according to the accuracy of the methods. “Table S1 and Figure S2” (line 275) – Here it must be pointed to the “Supporting information”. Table 4 indicates to Mz/Mw but the text to Mw/Mn – Why? “21.06%”, “17.45%”, “25.96” and “27.86%” (line 352/353) – Maximum one decimal place. Uniform use of lower case in all titles of the papers: References 7 (line 418(419), 11 (line 426), 16 (line 436/437), 27 (line 457/458), 34 (line 476/477), 36 (line 480/181), 42 (line 495/469), 44 (line499/500), 47 (line 507), line 49 (line 511), 61 (line 536/537), 62 (line 538), 63 (line 540/541). Replace “Makronwlekulare” with “Makromolekulare” (line 507).

Round 2

Reviewer 1 Report

Manuscript ID polymers-586511.R1

Authors modified some parts of the manuscript according the reviewers comments. However some points need to be carefully improved to avoid mistakes. In introduction section authors modified text and added the following idea “There are three methods for determination of molecular weight of cellulose: 1. Literature review.” Literature review is a method for MW determination? This need to be corrected and the authors should be more careful when writing and reviewing the manuscript before submission.

13C NMR chemical shifts are not in agreement with literature. See for example: Gustavsson, S.; Alves, L.; Lindman, B.; Topgaard, D. Polarization transfer solid-state NMR: a new method for studying cellulose dissolution. RSC Advances 2014, 4, 31836-31839, doi:10.1039/C4RA04415K. The measurements were referenced to some internal or external standard? If not please repeat the measurements or collect a reference spectra (ex. a-glycine) and correct the obtained spectra.

Line 348-350: Authors said: “As was shown in Figure 5, the crystallinity of the regenerated cellulose samples (210.6‰ for RC6 and 174.5‰ for RC7) apparently decreased compared with the original cellulose samples (259.6‰ for C6 and 278.6‰ for C7),…” Crystallinity is higher than 100%? Is that possible?

Line 361-363: “So, the variance in the [η]s and DPs was most likely due to the recrystallization of the original cellulose I into cellulose ΙΙ after regeneration from the solution”. Recrystallization can affect DP? How is that possible? If is a solubility problem, as stated by authors, the method is suitable for DP measurement?

A deep review is necessary to the manuscript and these basic ideas and concepts need always be present to avoid mistakes.

Reviewer 2 Report

Despite addition of lots of new data and editing old version, the cellulose samples used in this work were not well characterized and not defined perfectly. In new version also they didn't mentioned the cellulose samples purity.  

Reviewer 3 Report

The present contribution gives some new insight into how to determine the molecular weight of cellulose. After very minor revision of the revised version the paper should be now published in the journal “Polymers” without additional reviewing.

Subscript the second letters in the symbols for the molecular weights (line 306 – 308). Table 5: Non-zero decimal numbers places according to the accuracy of the methods.
